# METTL3 Promotes the Differentiation of Goat Skeletal Muscle Satellite Cells by Regulating MEF2C mRNA Stability in a m^6^A-Dependent Manner

**DOI:** 10.3390/ijms241814115

**Published:** 2023-09-14

**Authors:** Sen Zhao, Jiaxue Cao, Yanjin Sun, Helin Zhou, Qi Zhu, Dinghui Dai, Siyuan Zhan, Jiazhong Guo, Tao Zhong, Linjie Wang, Li Li, Hongping Zhang

**Affiliations:** 1Farm Animal Genetic Resources Exploration and Innovation Key Laboratory of Sichuan Province, Sichuan Agricultural University, Chengdu 611130, China; zhaosen97@126.com (S.Z.); jiaxuecao@sicau.edu.cn (J.C.); s18098064595@163.com (Y.S.); a152136157@163.com (H.Z.); 15296542810@139.com (Q.Z.); 71317@sicau.edu.cn (D.D.); siyuanzhan@sicau.edu.cn (S.Z.); jiazhong.guo@sicau.edu.cn (J.G.); zhongtao@sicau.edu.cn (T.Z.); wanglinjie@sicau.edu.cn (L.W.); 2Key Laboratory of Livestock and Poultry Multi-Omics, Ministry of Agriculture and Rural Affairs, College of Animal and Technology, Sichuan Agricultural University, Chengdu 611130, China

**Keywords:** RNA m^6^A methylation, METTL3, MEF2C, goat, MuSCs

## Abstract

The development of mammalian skeletal muscle is a highly complex process involving multiple molecular interactions. As a prevalent RNA modification, N6-methyladenosine (m^6^A) regulates the expression of target genes to affect mammalian development. Nevertheless, it remains unclear how m^6^A participates in the development of goat muscle. In this study, methyltransferase 3 (METTL3) was significantly enriched in goat longissimus dorsi (LD) tissue. In addition, the global m^6^A modification level and differentiation of skeletal muscle satellite cells (MuSCs) were regulated by METTL3. By performing mRNA-seq analysis, 8050 candidate genes exhibited significant changes in expression level after the knockdown of METTL3 in MuSCs. Additionally, methylated RNA immunoprecipitation sequencing (MeRIP-seq) illustrated that myocyte enhancer factor 2c (MEF2C) mRNA contained m^6^A modification. Further experiments demonstrated that METTL3 enhanced the differentiation of MuSCs by upregulating m^6^A levels and expression of MEF2C. Moreover, the m^6^A reader YTH N6-methyladenosine RNA binding protein C1 (YTHDC1) was bound and stabilized to *MEF2C* mRNA. The present study reveals that METTL3 enhances myogenic differentiation in MuSCs by regulating MEF2C and provides evidence of a post-transcriptional mechanism in the development of goat skeletal muscle.

## 1. Introduction

There is an increasing need for high-quality animal products as the world’s population continues to increase. Goat meat is rich in high-quality protein and unsaturated fatty acids, which help our body reduce the risk of cardiovascular disease [1,2]. Therefore, it is essential to explore the regulatory mechanisms underlying the regulation of muscle development in goats. Adults and domesticated animals generally weigh a considerable amount of their total weight in skeletal muscle, which is an essential body part [3]. It supports the body’s movement and fatigue relief [4,5]. Regarding human health, some muscle diseases (sarcopenia and muscular dystrophy) increase the risk of falls and fractures [2,6]. Notably, skeletal muscle is directly related to the production of meat. Myogenic regulatory factors (MRFs), paired box proteins (Paxs), myocyte enhancer factors (MEFs), and non-coding RNAs (ncRNAs) cooperate to sustain the development of animal skeletal muscle and are even involved in repair and regeneration after muscle injury [7,8]. Skeletal muscle satellite cells (MuSCs) are crucial in muscle regeneration after injury. When the body suffers an acute injury, MuSCs are activated for mitosis and further differentiate to myoblasts. Such a process can aid muscle regeneration and healing of damaged muscles [9].

Since the discovery of epigenetics, it has been widely reported that DNA methylation, histone modification, chromatin remodeling, and RNA modifications regulate skeletal muscle development [10,11]. N6-methyladenosine (m^6^A) is a prevalent RNA modification handling a variety of biological functions [12], including stem cell differentiation [13], development [14], disease [15], and sex determination [16]. Methyltransferase 3 (METTL3) and Methyltransferase 14 (METTL14) catalyze m^6^A, with METTL3 as the catalytic core [17]. Deletion of METTL3 induces embryonic lethality in mice [18]. When METTL3 was knocked out, the self-renewal capacity and m^6^A methylation of mouse embryonic stem cells (mESCs) were reduced [19]. Recent research illustrates that METTL3 is necessary for the development of skeletal muscle. It has also been shown that the knockdown of METTL3 inhibited the differentiation of C2C12 by reducing myogenic differentiation (*MyoD)* at post-transcriptional processes [11]. In addition, METTL3 regulates global m^6^A levels and inhibits myogenic differentiation in C2C12 [20]. A recent study revealed that METTL3 inhibits activin 2a receptor (ACVR2A) synthesis, mediating skeletal muscle hypertrophy [21]. Additionally, METTL3 regulates *tet methylcytosine dioxygenase 1 (TET1)* mRNA stability and the myogenic differentiation process in an m^6^A manner [22]. Nevertheless, METTL3 may exhibit different expression levels in different cell types [23]. Thus, it is still necessary to fully understand how METTL3 and m^6^A influence the development of goat muscle.

Here, we reveal the dynamic change and function of METTL3 in goat MuSCs. *METTL3* mRNA was highly enriched in postnatal goat muscle tissue. In particular, METTL3 was necessary for MuSCs myogenic differentiation and significantly regulated global m^6^A levels. Mechanistically, METTL3 promoted myocyte enhancer factor 2c (MEF2C) expression via m^6^A modification during myogenic differentiation, which depends on the protection of m^6^A reader YTH N6-methyladenosine RNA binding protein C1 (YTHDC1) on *MEF2C* mRNA stability. Our results unveil a post-transcriptional regulation mechanism and provide novel ideas on the role of m^6^A modification in the development of goat skeletal muscle.

## 2. Results

### 2.1. Homology Analysis of Goat METTL3 and Its Expression Profile in Tissues and MuSCs

To systematically understand the METTL3 protein, we analyzed the protein homology. According to the phylogenetic tree constructed by MEGA software [24], the goat (*Capra hircus*) METTL3 protein has a close homology to that of sheep (*Ovis aries*). Meanwhile, it is relatively distinct from the mouse (*Mus musculus*) and zebrafish (*Danio rerio*) METTL3 protein (Figure 1A). Using the SMART online tool [25], we analyzed the METTL3 domain in humans, mice, zebrafish, sheep, pigs, cattle, and goats. We found that METTL3 in all species mentioned above contained the MT-A70 domain (Figure 1B), which is the core region responsible for methyl transfer [17]. Based on the results of bioinformatics analysis, we hypothesized that the function of METTL3 was a methyltransferase in goats.

We performed qPCR to explore the spatio-temporal expression profile of METTL3 in goat tissues. Fetal development is vital for muscle development, forming the muscle fibers in the embryonic period [26]. Therefore, goat tissues at 105 embryonic days (E105) were selected for METTL3 expression assessment. *METTL3* transcripts were most enriched in the lung, followed by the longissimus dorsi (LD) muscle (Figure 1C). We further quantified METTL3 levels in LD tissue from goats aged between 90 embryonic days (E90) and 3 days after birth (B3) and found that transcripts of *METTL3* were gradually increased in the prenatal period and rose sharply at B3 (*p* < 0.05) (Figure 1D).

We successfully isolated and cultured MuSCs marked by Pax7 and the MyHC protein, respectively (Figure 1E). Compared with cells cultured in GM-3 (growth medium for 3 days), GM-5, and DM-2 (differentiation medium for 2 days), METTL3 was expressed at a higher level in DM-5 cells, a late stage of differentiation (*p* < 0.05) (Figure 1F). Intriguingly, the total m^6^A level initially reduced and then increased with the differentiation of MuSCs (Figure 1G). Moreover, compared with the control (siNC), small interfering RNA targeting METTL3 (siMETTL3) reduced METTL3 levels in goat MuSCs (Figure 1H and Appendix A), as well as the global m^6^A levels (*p* < 0.01) (Figure 1I). Our results suggest that METTL3 is functionally involved with skeletal muscle development in an m^6^A-dependent manner in goat MuSCs.

### 2.2. METTL3 Promotes Myogenic Differentiation in Goat MuSCs

To confirm the function of METTL3 in myogenic differentiation, we both silenced and overexpressed METTL3 in MuSCs to evaluate whether the METTL3 pathway drives MuSCs differentiation. As expected, qPCR and WB results showed that the knockdown of METTL3 (siMETTL3) significantly downregulated the myogenic differentiation (MyoD), myogenin (MyoG), and myosin heavy chain (MyHC) mRNA and MyHC protein expression level (*p* < 0.05) (Figure 2A,B). Meanwhile, the myotubes presented as MyHC immunofluorescence were rarely formed in siMETTL3-treated cells, and the fusion index was significantly retarded (*p* < 0.01) (Figure 2C).

On the contrary, ectopic METTL3 using a pEGFP-METTL3 overexpression vector (*p* < 0.01) (Figure 2D,E) promoted the expression of differentiation marker genes, including *MyoD* and *MyoG*. Consistently, the formation of myotubes was dramatically enhanced after METTL3 overexpression (*p* < 0.05) (Figure 2F,G). These imply that METTL3 is necessary for the differentiation of MuSCs.

### 2.3. mRNA-Seq and Methylated RNA Immunoprecipitation Sequencing (MeRIP-Seq) Identify MEF2C as an Underlying Target of METTL3

To systematically investigate the myogenic differentiation gene profile regulated by METTL3 in MuSCs, we performed mRNA-seq using METTL3-silenced cells. A total of 241, 219, and 162 clean reads were obtained with a mapping rate between 86.09 and 93.18% (Appendix A). The correlation and PCA analysis suggest that our experiments had good reproducibility of samples within the same group (Appendix A). Then, 8050 genes were defined as differentially expressed genes (DEGs) (Appendix A). Among these, six genes were randomly selected to evaluate the accuracy of the mRNA-seq data, which was confirmed by performing qPCR analysis (Appendix A). Further, GO analysis revealed that RNA metabolism and methyltransferase activity of MuSCs were affected by METTL3 knockdown (Figure 3A). Additionally, these DEGs were significantly enriched in 18 signaling pathways, including Notch, MAPK, RNA degradation, fatty acid metabolism, AMPK, Hedgehog, and focal adhesion, according to KEGG enrichment analysis (Figure 3B and Appendix A).

Among them, it was identified that the MAPK and Notch pathways are crucial for skeletal muscle growth [27,28]. Subsequently, the protein interaction networks of DEGs in MAPK and Notch pathways were constructed using the String tool and Cytoscape software [29,30]. MAPK3, MAPK14, AKT1, Notch3, Notch2, and MEF2C were important nodes related to muscle differentiation (Figure 3C). To mechanistically understand whether these METTL3-associated genes are related to m^6^A modification, we combined our previous MeRIP-seq data (LD muscle from embryonic 60 days (E60d) vs. birth 12 months (B12m)) and identified five genes (MYC, MEF2C, NOTCH2, HES1, and HEY1) showing a dynamic m^6^A modification pattern (Figure 3D and Appendix A). MeRIP-seq data indicated that the m^6^A peak was found on *MEF2C* mRNA (Figure 3E). Based on this, *MEF2C* was selected for further validation.

### 2.4. METTL3 Promotes Myogenic Differentiation of MuSCs by Increasing the Expression of MEF2C

We quantified and found that MEF2C dramatically increased as myogenic differentiation progressed (*p* < 0.001) (Appendix A), coinciding with the previous study [31]. Further, we investigated whether knockdown and overexpression of METTL3 regulated the expression of MEF2C by qPCR and WB analysis. Expectedly, the levels of *MEF2C* transcripts and protein were significantly decreased in METTL3-knockdown MuSCs (*p* < 0.05) (Figure 4A,B). Conversely, ectopic METTL3 dramatically elevated the expression of MEF2C mRNA and protein in METTL3-overexpressing cells (*p* < 0.05) (Figure 4C,D).

Additionally, when MEF2C was overexpressed, the levels of differentiation marker gene MyoD and MyoG were significantly promoted (*p* < 0.05 or *p* < 0.01) (Figure 4E,F). Moreover, the myotube fusion was significantly increased by ectopic MEF2C (*p* < 0.01) (Figure 4G). Nevertheless, overexpressing MEF2C failed to alter levels of METTL3 mRNA or protein (Figure 4E,F). Furthermore, we overexpressed MEF2C in siMETTL3-transfected cells and found that deficiency of METTL3 significantly inhibited the myogenic differentiation gene MyoD and MyHC, while the addition of MEF2C partly rescued these negative effects (Figure 4H). In general, METTL3 supports myogenic differentiation through MEF2C in MuSCs.

### 2.5. METTL3-m^6^A-YTHDC1 Stabilizes MEF2C Transcripts

To precisely define the m^6^A modification of MEF2C, we used the online tool SRAMP [32] and found that *MEF2C*-1728^A^ (XM_005683422.3 1728 A site) has the highest probability of m^6^A modification. (Figure 5A). Furthermore, we performed MeRIP-qPCR and found that the *MEF2C*-1728^A^ was enriched on m^6^A antibody, compared to IgG (*p* < 0.05) (Figure 5B). This result indicated that m^6^A modification occurred at the *MEF2C*-1728^A^. In addition, the MeRIP-qPCR assay demonstrated that the m^6^A level of MEF2C was reduced when METTL3 knockdown (*p* < 0.01) (Figure 5C). To confirm whether METTL3 regulates the expression of MEF2C in an m^6^A-dependent manner, we performed the luciferase reporter assay using wild-type (WT) (*MEF2C*-1728^A^) and m^6^A motif mutant (*MEF2C*-1728^T^, MUT) vectors. As expected, the knockdown of METTL3 or overexpressing METTL3 reduced or elevated luciferase activity (*p* < 0.05), respectively, while MUT activity was insignificantly changed (Figure 5D). The results show that METTL3 promotes the expression of MEF2C through upregulating m^6^A methylation modification at the *MEF2C*-1728^A^.

It is well-known that the m^6^A-RNAs are recognized by m^6^A ‘readers’, consequently impacting their metabolism [8]. We used the catRAPID Omics v2.0 online tool [33] and found that YTHDC1 owns a higher interaction propensity with *MEF2C* mRNA (Figure 5E and Appendix A). Then, we performed RIP assay and found that *MEF2C*-1728^A^ were significantly enriched in the YTHDC1 antibody (*p* < 0.05) (Figure 5F), suggesting that the YTHDC1 protein recognizes and binds to the m^6^A modification site of *MEF2C*. The expression pattern of *YTHDC1* was similar to that of METTL3 and MEF2C during MuSCs differentiation (*p* < 0.05) (Appendix A). When we successfully knocked down YTHDC1 using siYTHDC1 in MuSCs (Appendix A), MyHC and MEF2C levels were significantly suppressed (*p* < 0.05) (Figure 5G,H). Moreover, *MEF2C* mRNA exhibited a shorter half-life in YTHDC1-deficiency cells (*p* < 0.05) (Figure 5I). In summary, our results suggest that METTL3 cooperated with YTHDC1 to regulate myogenic differentiation by increasing the stability of *MEF2C* mRNA.

## 3. Discussion

The intricate process of muscle development involves the combined participation of several transcription factors, growth factors, non-coding RNAs, and related pathways [34]. However, the post-transcriptional mechanism influencing muscle development still needs to be better understood. Previous research has indicated that m^6^A is critical in determining RNA fate [35,36,37,38]. Abundant m^6^A modifications were identified in the human brain, lung, and muscle tissues [39]. This present study found that the expression of METTL3 indicated the highest level in the lung, followed by muscle and heart, which was indirectly considered conservative among different species in terms of m^6^A modification.

Studies have shown that METTL3 controls the global m^6^A levels in both porcine-induced pluripotent stem cells and mouse cardiac fibroblasts [40,41]. Additionally, METTL3 and m^6^A abundance increase during skeletal muscle hypertrophy in mice, and the deletion of METTL3 suppresses muscle cell growth [21]. Knockdown of METTL3 significantly inhibits myoblast differentiation and decreases the expression of MYH3 in bovine [42]. Our study found a similar trend in that the expression of METTL3 of goat MuSCs gradually increased with the time of muscle differentiation. Furthermore, the knockdown of METTL3 reduced the total RNA m^6^A level, indicating that METTL3 regulated the global m^6^A level in MuSCs during myogenic differentiation. Meanwhile, METTL3 promoted MuSCs differentiation and boosted differentiation marker genes. The above findings suggest that m^6^A modification promotes muscle development in goats.

Next, we found that DEGs were enriched to Notch and MAPK pathways by analyzing mRNA-seq data from siMETTL3 and siNC cells. It is well known that Notch and MAPK pathways are essential for MuSCs’s self-renewal and differentiation [43,44,45]. Furthermore, METTL3 regulates the proliferation of mouse muscle stem cells by the Notch pathway [46]. Another study reported that m^6^A modification mediated by the demethylase FTO regulates mTOR-PGC-1α and p38 MAPK signaling pathways, thereby regulating muscle development [47,48]. In this study, we determined some DEGs enriched in Notch and MAPK pathways, so we used those DEGs to construct a protein interaction network. Interestingly, some critical nodes were focused, such as MAPK3, MAPK14, AKT1, Notch3, Notch2, and MEF2C. However, whether METTL3 directly regulates the conduction of these pathways in an m^6^A-dependent manner in goat muscle development requires further study.

We determined that METTL3 promotes MEF2C’s mRNA and protein levels. Based on MeRIP-seq and mRNA-seq, we found that *MEF2C* was the potential target gene of METTL3 and m^6^A modification gene and verified this using qRT-PCR and MeRIP-PCR. MEF2C was reportedly involved with cardiomyocyte repair, neural development, skeletal muscle regeneration, and smooth muscle migration [49,50,51,52]. During skeletal muscle development, MEF2C regulates muscle integrity and myoblast differentiation [53,54]. Additionally, MEF2C cooperates with MyoD to control myogenic gene expression, supporting muscle development [55]. In this study, we found that the expression level of MEF2C was higher in the DM of MuSCs than in the GM and that MEF2C overexpression promoted MuSCs myogenic differentiation.

METTL3-regulated m^6^A affects the expression of target gene mRNAs. METTL3 knockdown increases YTHDF2-dependent mRNA stability and promotes the adipogenesis of porcine BMSCs by reducing the m^6^A level of *JAK* mRNA [47]. The m^6^A-reader IGF2BP3 stabilizes *RCC2* mRNA by recognizing m^6^A [56]. In our study, MeRIP-qPCR results showed that the *MEF2C*-1728^A^ fragment was significantly enriched with the m^6^A antibody. The deficiency of METTL3 in MuSCs considerably reduced the m^6^A level of *MEF2C*. Then, the luciferase assay further verified that this m^6^A site played a central role in the regulation process of METTL3. These results suggest that METTL3-dependent m^6^A regulates the expression of MEF2C in MuSCs.

According to reports, YTHDC1 is essential in regulating mRNA export and splicing in the nucleus [57]. Additionally, the interaction of METTL3 with chromatin that governs the integrity of IAP heterochromatin was improved by YTHDC1 [58]. In addition, the loss of YTHDC1 promoted the binding of target mRNA to RRP6 and decreased mRNA stability, revealing the underlying mechanism of YTHDC1 regulating mRNA stability [59]. In this study, we verified that *MEF2C* could bind to YTHDC1 by RIP experiments, and the expression of YTHDC1 was higher in the differentiation cells. Furthermore, the silence of YTHDC1 significantly decreased MyHC and MEF2C, as well as the half-life of *MEF2C* transcripts. Our results demonstrate that YTHDC1 recognizes the m^6^A modification of MEF2C and increases mRNA stability.

## 4. Materials and Methods

### 4.1. Animals and Samples Collection

The Jianyang Dageda farm (Jianyang, China) provided Jianzhou big-eared goats for use as experimental animals in this study. The Jianzhou big-eared goat is a breed of meat sheep bred by crossing the American Nubian goat with the native goat in Sichuan province, which is the second meat goat species after the Nanjiang Huang goat in China [60]. A standard diet (forage to concentrate ratio, 65:35) was provided to all healthy ewes (2–3 years old) twice daily, and they received free access to water in their stalls, followed by simultaneous estrus and artificial insemination treatments [61]. Goat fetuses and kids (females, *n* = 3) at 90 embryonic days (E90), 105 days (E105), 120 days (E120), 135 days (E135), and 3 days after birth (B3) were randomly selected (not related) and humanely sacrificed. Different tissues (LD, lung, heart, spleen, liver, and kidney) were sampled.

### 4.2. MuSCs Isolation and Identification

According to previous methods, the LD of the E90 goat (male) was successfully used to isolate the MuSCs for this study [62]. Then, we used the antibody against myogenic marker genes Pax7 (Santa Cruz, CA, USA) and MyHC (myosin heavy chain, Santa Cruz, CA, USA) for immunofluorescence. MuSCs were stored in liquid nitrogen tanks.

### 4.3. Phylogenetic Tree Construction and Protein Domain Analysis

The neighbor-joining statistical method of MEGA software (version 11; https://megasoftware.net; (accessed on 7 January 2023)) was used to draw a METTL3 phylogenetic tree [24]. The phylogenetic tree was constructed by the online tools of EvolView (https://evolgenius.info//evolview-v2/#login; (accessed on 6 March 2023)) [63]. The SMART online tool was used to analyze the METTL3 protein domain [25]. The NCBI database was used to obtain the METTL3 amino acid sequences in several species.

### 4.4. Cell Culture and Transfection

MuSCs were cultured at 5% CO_2_ and 37 °C in GM (growth medium) containing 89% DMEM, 10% FBS (Gibco, Grand Island, NY, USA), and 1% penicillin-streptomycin (Invitrogen, Bohemia, NY, USA). The DM (differentiation medium) containing 98% DMEM and 2% horse serum (Gibco) was changed daily to promote myogenic differentiation after the MuSCs density reached 80% to 90%. Plasmids and siRNAs were transfected into MuSCs using Lipofectamine 3000 (Life Technologies, Waltham, MA, USA).

### 4.5. Gene silencing and Plasmid Construction

RiboBio (Guangzhou, China) designed and synthesized the siRNAs used for this experiment, and the siRNA sequences are shown in Appendix A. The CDS sequences of the *MEF2C* (NM_001314204.1) gene were amplified with specific primers, and the full-length of CDS was inserted into the pEGFP-N1 (Promega, Madison, WI, USA) vector by the homologous recombinant cloning kit (Vazyme, Nanjing, China) to construct overexpression plasmids. The METTL3 overexpression vector was synthesized by Hangzhou Youkang Biotechnology (Hangzhou, China). Appendix A shows the cloning primers.

### 4.6. Total RNA Isolation and qPCR

Total RNA was isolated from tissues and MuSCs using RNAiso Plus (Takara, Dalian, China). The cDNAs were obtained by the PrimeScript™ RT kit (Takara). In addition, SYBR Premix Ex TaqTM II (Takara) was used for qPCR. Using the GAPDH and 2^−ΔΔCt^ method to normalize relative RNA expression [64]. All primers were detailed in Appendix A.

### 4.7. Immunofluorescence Analysis

MuSCs were washed thrice with PBS (5 min per session) and fixed with 4% paraformaldehyde. Then, 0.5% TritonX-100 was used to permeabilize the cells. Subsequently, the cells were incubated with primary antibodies (1:200, 4 °C, 12 h) and secondary antibodies (1:200, 37 °C, 2 h). Finally, ImageJ software (version 1.53; https://imagej.nih.gov/ij/; (accessed on 24 September 2021)) was used to count the fusion index [65]. Antibodies are as follows: anti-Pax7 (1:200, abs124153, absin), anti-MyHC (1:300, sc-378137, Santa Cruz, CA, USA), Cy3 Goat Anti-Mouse IgG (1:200, AS008, ABclonal, Wuhan, China), and FITC Goat Anti-Rabbit IgG (AS011, 1:200, ABclonal, Wuhan, China).

### 4.8. Western Blot Analysis

MuSC proteins were obtained by the total protein extraction kit (Beyotime, Shanghai, China). Proteins were denatured (100 °C, 5 min) by adding protein loading buffer (5× SDS-PAGE, Beyotime). ImageJ software (version 1.53; https://imagej.nih.gov/ij/; (accessed on 24 September 2021)) was used to perform the semi-quantitative study of proteins. The WB used the following antibodies: anti-β-tubulin (1:1000, 250007, ZENBIO, Chengdu, China), anti-MyHC (1:1000, sc-378137, Santa Cruz, Dallas, TX, USA), anti-MEF2C (1:500, 10056-1-AP, Proteintech, Wuhan, China), anti-YTHDC1 (1:1000, ab220159, Abcam, Cambridge, MA, USA), anti-METTL3 (1:1000, A8370, ABclonal, Wuhan, China), MyoD (1:1000, 18943-1-AP, Proteintech, Proteintech, Wuhan, China), anti-Mouse IgG (1:5000, 511103, ZENBIO, Chengdu, China), and anti- Rabbit IgG (AS014, 1:5000, ABclonal, Wuhan, China).

### 4.9. Luciferase Reporter Assays

The wild-type (WT) and m^6^A motif mutant (A to T mutation at 1728 site, MUT) of the *MEF2C*-1728 region were inserted into the psiCHECK-2 vector. The WT and MUT vectors were transfected to pEGFP-METTL3 and siMETTL3-transfected MuSCs. The luciferase activity was quantified by the Double-Luciferase Reporter Assay Kit (Transgen, Beijing, China). The MUT vectors were synthesized by Tsingke (Beijing, China). These primers are shown in Appendix A.

### 4.10. Total m^6^A Modification Level Analysis

Total RNA was isolated from MuSCs in the proliferative phase (GM-3, GM-5) and differentiation phase (DM-2, DM-5) using RNAiso Plus. The global m^6^A methylation level of RNA was detected using the m^6^A RNA Methylation Quantification Kit (Epigentek, Farmingdale, NY, USA).

### 4.11. RNA Immunoprecipitation (RIP) Assay

MuSCs were collected and lysed for RIP assays to assess the binding ability of YTHDC1 and MEF2C. Subsequently, the Magna RIP™ RNA-Binding Protein Immunoprecipitation Kit (Sigma-Aldrich, St. Louis, MO, USA) was used for RIP assays. These antibodies used in the RIP assay were YTHDC1 (1:20, 77422s, Cell Signaling Technology, Danvers, MA, USA) and IgG (Sigma-Aldrich, St. Louis, MO, USA).

### 4.12. RNA Stability Assays

Actinomycin D (AcTD, A1410, Sigma-Aldrich) was used on YTHDC1 knockdown or control MuSCs for 0 h, 1 h, 2 h, and 4 h to inhibit global mRNA transcription [66].

### 4.13. Methylated RNA Immunoprecipitation (MeRIP)

MuSCs were used to obtain m^6^A mRNA using the Magna MeRIP^TM^ m^6^A Kit (Sigma-Aldrich).

### 4.14. mRNA-Seq and Bioinformatics Analysis

Novogene (Beijing, China) performed the library preparation and mRNA-seq. Briefly, MuSCs were cultured until day 3 of differentiation and transfected with siNC or siMETTL3 into cells (*n* = 3). Total RNA was isolated using RNA iso Plus. The Agilent 2100 bioanalyzer and agarose gel electrophoresis (1%) were used to evaluate the integrity of RNA samples (Appendix A). The NEBNext^®^ Ultra™ RNA Library Prep Kit for Illumina^®^ (NEB, Beverly, MA, USA) was used for building the library. In addition, the Illumina HiSeq 2500 (Illumina, San Diego, CA, USA) was used to sequence (2 × 150 bp pair-end) qualified RNA.

Next, six mRNA-seq libraries (3 siNC and 3 siMETTL3 samples) generated 261,101,486 raw reads. Clean reads were obtained from raw reads, as described in previous studies [67]. Moreover, the Q30 value of all libraries was more than 92.39% (Appendix A). Hisat2 mapped clean reads to the goat ARS1 reference genome (GCF_001704415.1) [68]. The DESeq2 R Package was used to carry out differential read count analysis for identifying differentially expressed genes (DEGs, | log2 (FoldChange) | > 0, and padj = 0.05) [69]. Finally, the clusterProfile package was used for GO and KEGG enrichment analysis, and padj < 0.05 was considered significantly enriched [70].

### 4.15. Bioinformatics Analysis of Networks and Genes

Protein interaction networks were constructed by String tool (https://cn.string-db.org (accessed on 6 January 2023)) and Cytoscape software (version 3.9.1; https://metascape.org/g-p/index.html#/main/step1 (accessed on 8 January 2023)) [29,30]. The m^6^A site of *MEF2C* mRNA was predicted by the online tool SRAMP (http://www.cuilab.cn/sramp (accessed on 18 February 2023)) [32]. Subsequently, the binding potential of YTHDC1 to MEF2C was predicted by catRAPID (http://service.tart-aglialab.com (accessed on 22 February 2023)) [33].

### 4.16. Statistical Analysis

At least three biological replicates were performed to ensure the experiment’s reproducibility. An unpaired two-tailed Student’s *t*-test and one-way ANOVA with Tukey’s correction were used for two groups and multiple-group comparisons by GraphPad Prism (Version 9.0; https://www.graphpad.com (accessed on 16 August 2021)). All results were presented as mean ± SEM. Significance was denoted as * *p* ≤ 0.05.

## 5. Conclusions

In conclusion, our findings demonstrate that METTL3 expression regulates the global m^6^A level in muscle and is critical to the process of MuSCs differentiation. Moreover, METTL3 promotes the differentiation of MuSCs by regulating the expression of MEF2C. Mechanistically, METTL3 directly regulates the m^6^A modification of *MEF2C* mRNA, which is recognized by YTHDC1 and protects the stability of mRNA, thereby promoting MuSCs myogenic differentiation (Figure 6).

## Figures and Tables

**Figure 1 ijms-24-14115-f001:**
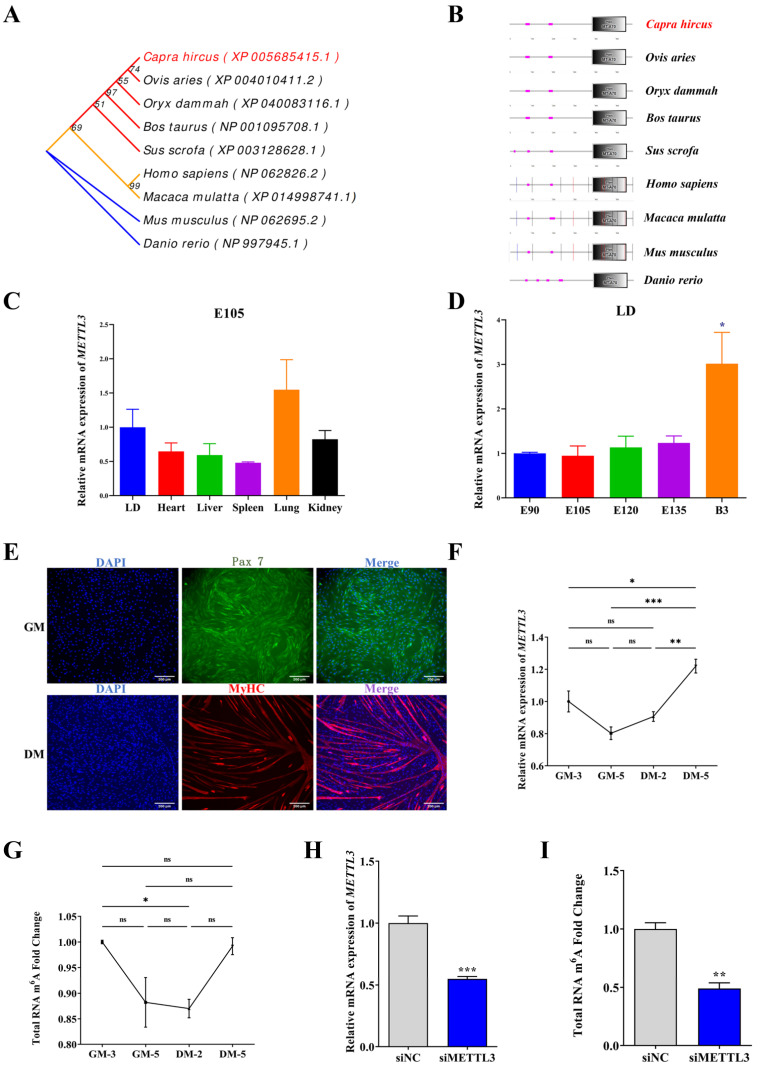
Characteristics and methylation transfer effects of goat METTL3. (**A**) Phylogenetic tree of METTL3. (**B**) Analysis of METTL3 protein domains. (**C**) qPCR analysis of METTL3 expression in different tissues of embryonic 105 days (E105) goats. (**D**) METTL3 expression in the longissimus dorsi (LD) tissue at different stages was measured using qPCR. (**E**) Pax7 (green) and MyHC (red) immunofluorescence staining were performed in MuSCs (scale bar: 200 μm). (**F**) METTL3 expression was determined by qPCR in the growth medium (GM) and differentiation medium (DM) of MuSCs. (**G**) Total RNA m^6^A levels in goat MuSCs at different stages were detected by methylation colorimetry. (**H**) qPCR analysis verified the knockdown of METTL3 in MuSCs. (**I**) Total RNA m^6^A levels in goat MuSCs were detected after transfection of siMETTL3 using methylation colorimetry. Results are represented as the mean ± SEM, * *p* < 0.05, ** *p* < 0.01, and *** *p* < 0.001, and ns indicates no significance.

**Figure 2 ijms-24-14115-f002:**
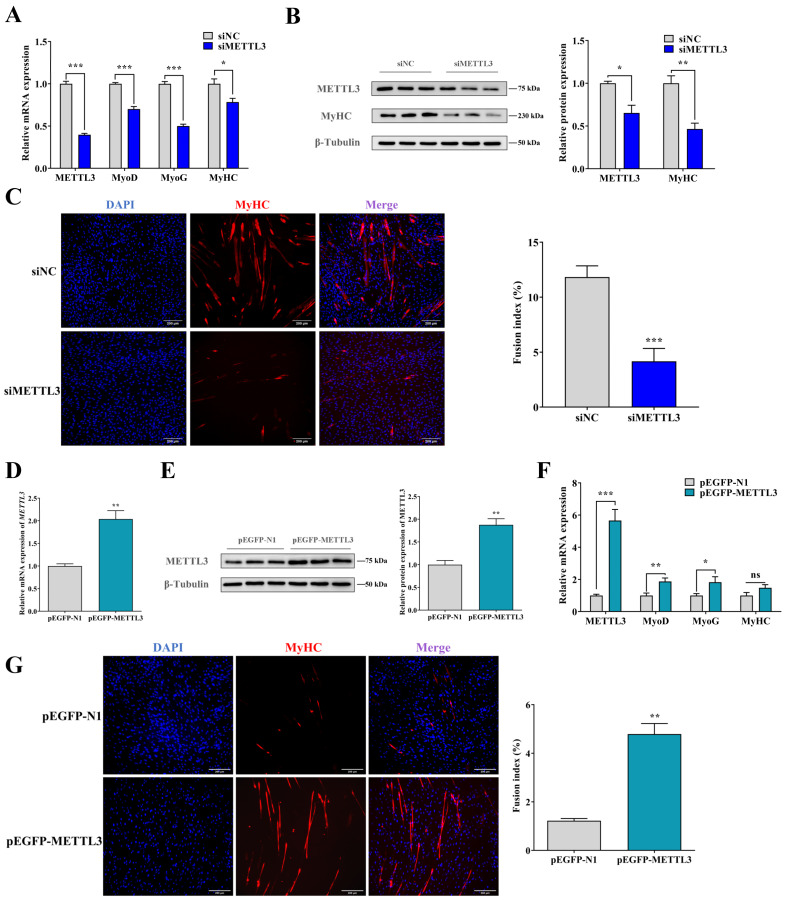
METTL3 expression promotes the differentiation of MuSCs. (**A**) METTL3, MyoD, MyoG, and MyHC expression after the knockdown of METTL3 by qPCR. (**B**) METTL3 and MyHC protein expression were measured by Western Blot analysis (WB) in METTL3-knockdown MuSCs. (**C**) The formation of myotubes was detected using the MyHC (red) immunofluorescence analysis after the knockdown of METTL3 (scale bar: 200 μm). (**D**,**E**) METTL3 mRNA (**D**) and protein (**E**) expression were detected after transfecting pEGFP-METTL3. (**F**) qPCR analysis of METTL3, MyoD, MyoG, and MyHC expression after transfection of pEGFP-METTL3. (**G**) The formation of myotubes was detected using the MyHC (red) immunofluorescence analysis in pEGFP-METTL3-transfected cells (scale bar: 200 μm). Results are represented as the mean ± SEM, * *p* < 0.05, ** *p* < 0.01, *** *p* < 0.001, and ns indicates no significance.

**Figure 3 ijms-24-14115-f003:**
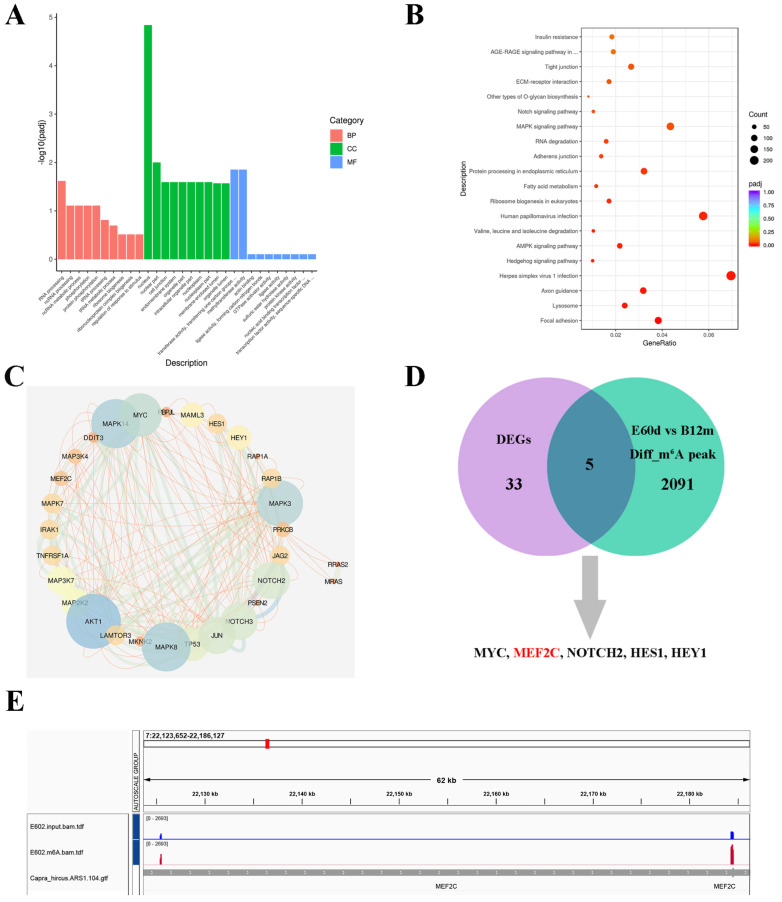
MEF2C is a potential target of METTL3 in goat MuSCs. (**A**) GO analysis of all differentially expressed genes (DEGs). (**B**) The top 20 enriched KEGG signaling pathways of all DEGs. (**C**) The String tool and the Cytoscape software were used for constructing the DEGs protein interaction network of the MAPK and Notch signaling pathways. (**D**) DEGs were analyzed by combining MeRIP-seq (E60d vs. B12m) and mRNA-seq data. (**E**) The m^6^A abundances in *MEF2C* transcript in goat E60d skeletal muscle as detected by MeRIP-seq.

**Figure 4 ijms-24-14115-f004:**
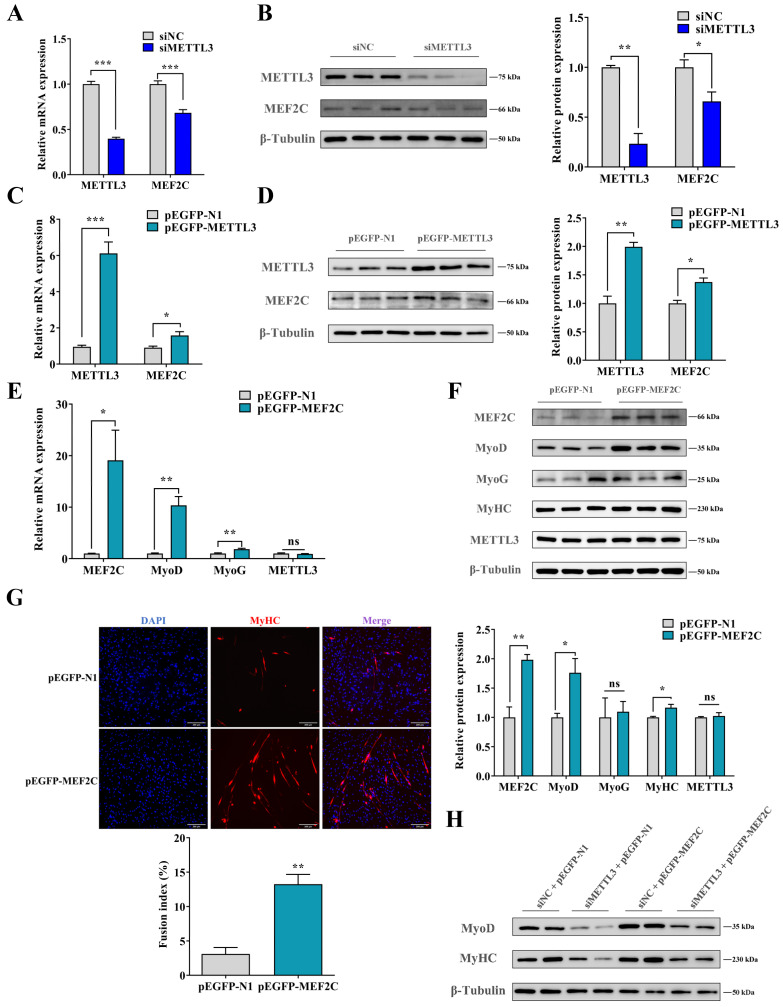
METTL3 promotes the differentiation of MuSCs by upregulating the MEF2C expression. (**A**,**B**) METTL3 and MEF2C mRNA (**A**) and protein (**B**) expression were measured after the knockdown of METTL3. (**C**,**D**) METTL3 and MEF2C mRNA (**C**) and protein (**D**) expression were measured after transfecting pEGFP-METTL3. (**E**,**F**) MEF2C, MyoD, MyoG, and METTL3 mRNA (**E**) and protein (**F**) expression were detected after transfecting pEGFP-MEF2C. (**G**) The formation of myotubes was detected using the MyHC (red) immunofluorescence analysis after transfecting pEGFP-MEF2C (scale bar: 200 μm). (**H**) siMETTL3 and pEGFP-MEF2C were co-transfected into MuSCs, and MyoD and MyHC protein expression were measured by WB analysis. Results are represented as the mean ± SEM, * *p* < 0.05, ** *p* < 0.01, *** *p* < 0.001, and ns indicates no significance.

**Figure 5 ijms-24-14115-f005:**
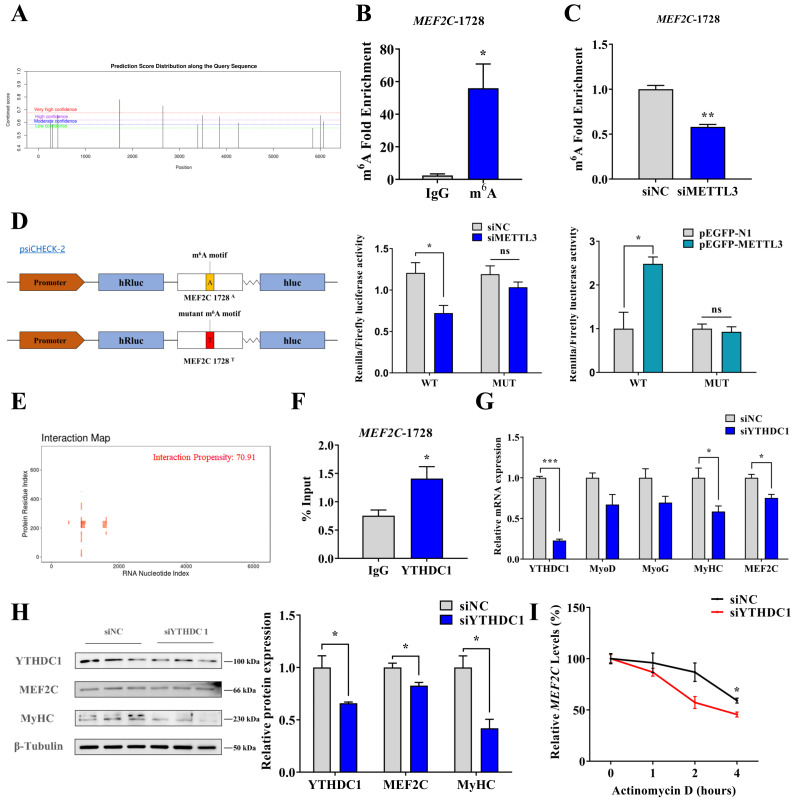
The METTL3-m^6^A-YTHDC1 axis promotes *MEF2C* mRNA stability during the myogenic differentiation of MuSCs. (**A**) The specific sites of m^6^A methylation modification in *MEF2C* (XM_005683422.3) mRNA were predicted by SRAMP. (**B**) MeRIP-qPCR was used to assess whether *MEF2C*-1728 had m^6^A modification. (**C**) MeRIP-qPCR measured the m^6^A level of *MEF2C*-1728 after the METTL3 knockdown. (**D**) Luciferase assay for silencing or overexpression of METTL3. (**E**) The potential of YTHDC1 to bind to *MEF2C* mRNA was predicted using the catRAPID. (**F**) RIP-qPCR analysis was carried out using anti-YTHDC1 (IP) or anti-IgG (negative control). (**G**) Expression of YTHDC1, MyoD, MyoG, MyHC, and MEF2C by qPCR analysis in YTHDC1-knockdown MuSCs. (**H**) YTHDC1, MyHC, and MEF2C protein expression were determined by WB analysis after transfecting siYTHDC1. (**I**) YTHDC1 knockdown or control MuSCs were treated with Actinomycin D (AcTD) and qPCR analysis of YTHDC1 expression. Results are represented as the mean ± SEM, * *p* < 0.05, ** *p* < 0.01, *** *p* < 0.001, and ns indicates no significance.

**Figure 6 ijms-24-14115-f006:**
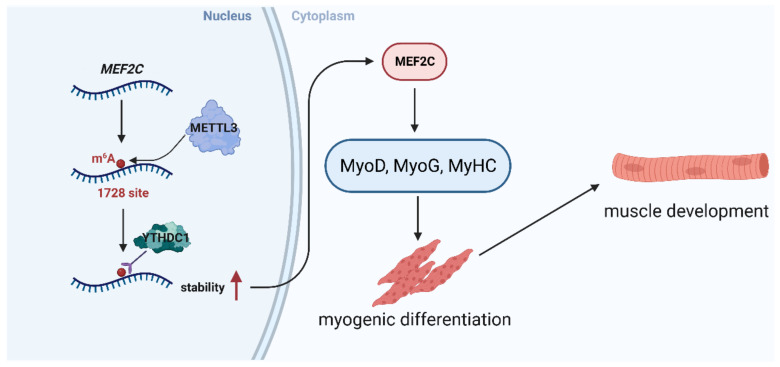
BioRender was used for creating a schematic illustration of METTL3 regulating MuSCs myogenic development. Red arrows indicate upregulation of stability.

## Data Availability

The data from the current study are exhibited in the manuscript and Appendix A.

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
