# Peer review of "METTL3 Promotes the Differentiation of Goat Skeletal Muscle Satellite Cells by Regulating MEF2C mRNA Stability in a m6A-Dependent Manner"

_ijms, 2023, doi:10.3390/ijms241814115_

Round 1

Reviewer 1 Report

Dear Authors,

in my opinion your work is well presented and it is robust both for experimental design and for the promising obtained reults. I suggest to extend the analysis and related speculations, in the future, also on other mammals and animal species.

Below only some littel observations and comments.

Please, define all the acronyms in the abstract and along the text, also at figures legend and caption, even if there is the Abbreviation sections.

Line 50: please modify as follows "...that METTL3 is necessary"

Line 89; please modify as follows "....combining with METTL3 level"

Line 233: please modify as follows: "......METTL3 controls the global...."

I suggest to implement the paragrapf relative to molecular analysis, adding important details for RTqPCR assay.

Please check the references style and format, according to IJMS Journal guideline.

Best regards.

English language is good, but I suggest a further overall check.

Author Response

Response to Reviewer 1 Comments

Dear Reviewer:

We feel great thanks for your professional review work on our manuscript entitled “METTL3 promotes the differentiation of goat skeletal muscle satellite cells by regulating MEF2C mRNA stability in m6A-dependent manner” (Manuscript ID: ijms-2576096). As you are concerned, there are several problems that need to be addressed. The comments were all very insightful and helpful for revising and improving our paper, and served as clear guidelines. We have carefully studied the comments and made corrections, which we hope are met with your approval. The revised portions are marked in red in this version of the paper. The main corrections in the paper in response to the reviewers’ comments are as following:

Point 1: Please, define all the acronyms in the abstract and along the text, also at figures legend and caption, even if there is the Abbreviation sections.

Response 1: Thanks for your comments. We have added definitions of acronyms to the abstract and along the text. For some important genes, we also added the full names of genes. In addition, Based on your comments, we have repeated definitions of acronyms in figure legends and captions.

Point 2: Line 50: please modify as follows "...that METTL3 is necessary". Line 89; please modify as follows "....combining with METTL3 level". Line 233: please modify as follows: "......METTL3 controls the global....".

Response 2: Thanks for your comments. Your suggestions will make our sentences more clear. We have modified Line 50, Line 89, and line 233 according to your request.

Point 3: I suggest to implement the paragrapf relative to molecular analysis, adding important details for RTqPCR assay.

Response 3: Thanks for your comments. In this version of the manuscript, we have added the description of RT-qPCR detection of mRNA expression. For example:

Line 87: “We performed qPCR to explored the spatio-temporal expression profile of METTL3 in goat tissues. ”

Line 119: “As expected, qPCR and WB results showed that the knockdown of METTL3 (siMETTL3) significantly downregulated the myogenic dif-ferentiation (MyoD), myogenin (MyoG), and myosin heavy chain (MyHC) mRNA and MyHC protein expression level. ”

Line 147: “Then, 8050 genes were defined as differentially expressed genes (DEGs) (Fig. S3C and Table S6), among which six genes were randomly selected, and the accuracy of the mRNA-seq data was confirmed by qPCR analysis. ”

Line 176: “we investigated whether knockdown and overexpression of METTL3 regulated the ex-pression of MEF2C by qPCR and WB analysis.”

Point 4: Please check the references style and format, according to IJMS Journal guideline.

Response 4: Thanks for your comments. We are very sorry for our negligence. We have checked and modified the style and format of the references according to the IJMS Journal Guidelines.

Thank you again for your affirmation of the manuscript. We tried our best to improve the manuscript and made some changes to the manuscript. These changes will not influence the core content and framework of the paper.

We sincerely appreciate for your thorough and insightful work, and hope that the corrections will meet with approval.

Once again, thank you very much for your comments and suggestions.

Best regards.

Yours sincerely,

Sen Zhao

Reviewer 2 Report

Dear authors,

The manuscript tries to discover the function of METTL3 in differentiation of goat skeletal muscle. The manuscript is well written and structured, but several changes are necessary. The introduction provides sufficient backgroung, However, the authors does not indicate why the study carried out in goat and not in other species, and the significance in goat. The material and methods could be more explain. For example, the authors indicate that four animals has been used, but in the rest of MM section, only one animal shown, concretely, the animal E90 to generate MuSCs. In the line 324, the authors indicate results of table S1, but this table does not exist. However, tables S6, S7 and S8 are not named in the text. Finally, some information is missing from the results. For example, data shown in figure 1B is not explained in detail, or the study of expression of METTL3 transcripts in different organs (figure 1C) only wass carried out in the E105 animal, but the authors do not explain why only in this animal, or how they carried out that.

I believe that the manuscript should be carefully reviewed by the authors and they should include all the missing information before being re-evaluated for possible publication.

Author Response

Response to Reviewer 2 Comments

Dear Reviewer:

Thank you for your comments concerning our manuscript entitled “METTL3 promotes the differentiation of goat skeletal muscle satellite cells by regulating MEF2C mRNA stability in m6A-dependent manner” (Manuscript ID: ijms-2576096). The comments were all very insightful and helpful for revising and improving our paper, and served as clear guidelines. We have carefully considered the comments and have made revisions to the manuscript point by point. All the modifications are highlighted in red in the revised manuscript. Please check the detailed responses and explanations listed below.

Point 1: The introduction provides sufficient backgroung, However, the authors does not indicate why the study carried out in goat and not in other species, and the significance in goat.

Response 1: Thanks for your comments. We are very sorry for our negligence, we have added in the introduction section the importance of studying muscle development in goats. Line 33: “There is an increasing need for high-quality animal products as the world's population continues to increases. Goat meat is rich in high-quality protein and unsaturated fatty acids, which helps our body to reduce the risk of cardiovascular disease [1, 2]. Therefore, it is essential to explore the regulatory mechanisms underlying the regulation of muscle development in goats.”

Point 2: The material and methods could be more explain. For example, the authors indicate that four animals has been used, but in the rest of MM section, only one animal shown, concretely, the animal E90 to generate MuSCs.

Thanks for your comments. Regarding the question of cells, we isolated and purified the cells at four time points, which were embryonic day 90 (E90), embryonic day 105 (E105), embryonic day 120 (E120), and embryonic day 135 (E135), respectively. Then, we tested their differentiation potential according to the protocol reported before [3]. Among four samples, E90 MuSCs (Skeletal muscle satellite cells) showed the highest differentiation ability (Figure 1E). Therefore, we use the E90 cells for subsequent in vitro experiments.

Point 3: In the line 324, the authors indicate results of table S1, but this table does not exist. However, tables S6, S7 and S8 are not named in the text.

Response 3: Sorry for the mistake had been made when submitting. We have reattached Table S1 in the manuscript. The details of Table S1 are shown on page 8 of the Word document called “Supplementary Material”. In addition, we have marked Tables S6, S7, and S8 in red in the manuscript for your convenience.

Point 4: Finally, some information is missing from the results. For example, data shown in figure 1B is not explained in detail, or the study of expression of METTL3 transcripts in different organs (figure 1C) only wass carried out in the E105 animal, but the authors do not explain why only in this animal, or how they carried out that.

Response 4: Thanks for your comments. Your suggestion will make our article clearer. We have redescribed Figure 1B in the manuscript. Line 82: "Using the SMART online tool[4], we analyzed the METTL3 domain in human, mice, zebrafish, sheep, pig, cattle, and goats. We found that METTL3 in all species mentioned above contained the MT-A70 domain (Fig. 1B), which is the core region responsible for the methyl transfer [5]. Based on the results of bioinformatics analysis, we hypothesized that the function of METTL3 was a methyltransferase in goats."

Due to our oversight, we did not elaborate on the reasons for choosing E105 animals, and we would like to apologize again. We choose goats of this age for two main reasons. First of all, previous studies have indicated that deletion of METTL3 induces embryonic lethality in mice (Line 54 of the manuscript) [6]. In addition, when METTL3 was knocked out, the self-renewal capacity and m6A methylation of mouse embryonic stem cells (mESCs) were reduced (Line 54 of the manuscript) [7]. Therefore, METTL3 has an essential function in the embryonic stage of animals. In addition, we have added to the manuscript the importance of development during the embryonic period for muscle formation. Line 88: “Fetal development is vital for muscle development, forming the muscle fibers in the embryonic period[8]. We collected goat tissues from four embryonic stages for the above two reasons. Subsequently, to initially evaluate the function of METTL3, a mid-embryonic time point (E105) was selected for tissue expression profiling. Further, We focused on its expression in muscle tissue. Then, METTL3 expression was measured in different times of muscle tissue by qPCR.

In response to your question about how to measure METTL3 expression in tissues, we have a detailed description in Material Methods. Firstly, goat tissue samples were collected (Line 311 of the manuscript). Secondly, total RNA was extracted and reverse transcribed (Line 350 of the manuscript). Finally, the relative expression level of METTL3 mRNA in tissues was detected by qPCR (Line 351 of the manuscript).

Thank you again for your affirmation of the manuscript. We tried our best to improve the manuscript and made some changes to the manuscript. We sincerely appreciate for your thorough and insightful work, and hope that the corrections will meet with approval.

Once again, thank you very much for your comments and suggestions.

Best regards.

Yours sincerely,

Sen Zhao

References

  1. de Sousa, S. V.; Diogenes, L. V.; Oliveira, R. L.; Souza, M. N. S.; Mazza, P. H. S.; da Silva Júnior, J. M.; Pereira, E. S.; Parente, M. O. M.; Araújo, M. J.; de Oliveira, J. P. F., et al. Effect of dietary buriti oil on the quality, fatty acid profile and sensorial attributes of lamb meat. Meat Sci 2022, 186, 108734, https://doi.org/10.1016/j.meatsci.2022.108734.
  2. Cruz-Jentoft, A. J.; Bahat, G.; Bauer, J.; Boirie, Y.; Bruyère, O.; Cederholm, T.; Cooper, C.; Landi, F.; Rolland, Y.; Sayer, A. A., et al. Sarcopenia: revised European consensus on definition and diagnosis. Age and Ageing 2019, 48, 16-31, https://doi.org/10.1093/ageing/afy169.
  3. Zhao, W.; Chen, L.; Zhong, T.; Wang, L.; Guo, J.; Dong, Y.; Feng, J.; Song, T.; Li, L.; Zhang, H. The differential proliferation and differentiation ability of skeletal muscle satellite cell in Boer and Nanjiang brown goats. Small Ruminant Research 2018, 169, 99-107, https://doi.org/10.1016/j.smallrumres.2018.07.006.
  4. Letunic, I.; Doerks, T.; Bork, P. SMART 6: recent updates and new developments. Nucleic Acids Res 2009, 37, D229-32, https://doi.org/10.1093/nar/gkn808.
  5. Wang, X.; Feng, J.; Xue, Y.; Guan, Z.; Zhang, D.; Liu, Z.; Gong, Z.; Wang, Q.; Huang, J.; Tang, C., et al. Structural basis of N(6)-adenosine methylation by the METTL3-METTL14 complex. Nature 2016, 534, 575-8, https://doi.org/10.1038/nature18298.
  6. Geula, S.; Moshitch-Moshkovitz, S.; Dominissini, D.; Mansour, A. A.; Kol, N.; Salmon-Divon, M.; Hershkovitz, V.; Peer, E.; Mor, N.; Manor, Y. S., et al. Stem cells. m6A mRNA methylation facilitates resolution of naïve pluripotency toward differentiation. Science (New York, N.Y.) 2015, 347, 1002-6, https://doi.org/10.1126/science.1261417.
  7. Wang, Y.; Li, Y.; Toth, J. I.; Petroski, M. D.; Zhang, Z.; Zhao, J. C. N6-methyladenosine modification destabilizes developmental regulators in embryonic stem cells. Nat Cell Biol 2014, 16, 191-8, https://doi.org/10.1038/ncb2902.
  8. Du, M.; Yan, X.; Tong, J. F.; Zhao, J.; Zhu, M. J. Maternal obesity, inflammation, and fetal skeletal muscle development. Biol Reprod 2010, 82, https://doi.org/10.1095/biolreprod.109.077099.

Reviewer 3 Report

I have several two major questions.

1) What was the novel finding from this study? Were the findings from this study the well-known facts from previous studies using other species? Or what was the goat-specific results?

2) This study used fetuses and kids. There seems no information on the feeding conditions and stress managements (such as heat) for both the dam and offspring. Furthermore, there was no information on pedigree/family information.

Author Response

Response to Reviewer 3 Comments

Dear Reviewer:

Thank you for your comments concerning our manuscript entitled “METTL3 promotes the differentiation of goat skeletal muscle satellite cells by regulating MEF2C mRNA stability in m6A-dependent manner” (Manuscript ID: ijms-2576096). The comments were all very insightful and helpful for revising and improving our paper, and served as clear guidelines. We have carefully studied the comments and made corrections, which we hope are met with your approval. The revised portions are marked in red in this version of the paper. The main corrections in the paper in response to the reviewers’ comments are as following:

Point 1: What was the novel finding from this study?  Were the findings from this study the well-known facts from previous studies using other species?  Or what was the goat-specific results?

Response 1:

Thanks for your comments. In this study, a series of experiments were performed to determine the expression pattern and function of the methyltransferase METTL3 in goat muscle (Results 2.1 and 2.2). At the same time, we found that METTL3 is necessary for MuSCs myogenic differentiation and significantly regulates global m6A levels. In addition, we found a new pathway for METTL3 to regulate myogenic differentiation: METTL3 regulates MuSCs differentiation by affecting the expression of myogenic transcription factor MEF2C through the m6A axis (Line 223 of the manuscript).

About METTL3 studies on other species, based on your suggustion, we had stressed this part in the manuscript as following. Line 57 of the manuscript: "METTL3 inhibited the differentiation of C2C12 by reducing the myogenic differentiation (MyoD) at post-transcriptional processes[1]." Line 62 of the manuscript: "METTL3 regulates tet methylcytosine dioxygenase 1 (TET1) mRNA stability and the myogenic differentiation process in an m6A manner [2]. " Line 248 of the manuscript: "METTL3 and m6A abundance increase during skeletal muscle hypertrophy in mice, and the deletion of METTL3 suppressed muscle cell growth [3]. In addition, we added METTL3's role in bovine skeletal muscle in the manuscript. Line 250 of the manuscript: Knockdown of METTL3 significantly inhibited myoblast differentiation and decreased the expression of MYH3 in bovine[4].

For the goat-specific results, according to your comments, we had described it in manuscript. In our study, we found an unreported result in goats: METTL3 directly regulates the m6A modification of MEF2C mRNA, which is recognized by YTHDC1 and protects the stability of mRNA, thereby promoting myogenic differentiation of MuSCs (Line 304 of the manuscript).

Point 2: This study used fetuses and kids. There seems no information on the feeding conditions and stress managements (such as heat) for both the dam and offspring.  Furthermore, there was no information on pedigree/family information.

Response 2: Thank you again for your constructive comments during the review process. We have added managed rearing conditions and pedigree for animals to the Materials and Methods section. Line 313 of the manuscript: “he Jianzhou Big-eared Goat is a breed of meat sheep bred by crossing the American Nubian goat with the native goat in Sichuan province, which is the second meat goat species after the Nanjiang Huang Goat in China[5]. A standard diet (forage to concentrate ratio, 65:35) will be provided to all healthy ewes (2–3 years old) twice daily, and they will receive free access to water in their stalls, followed by simultaneous estrus and artificial insemination treatments[6]. Goat fetuses and kids (females, n=3) at embryonic 90 days (E90), 105 days (E105), 120 days (E120), 135 days (E135), and three days after birth (B3) were randomly selected (not related) and humanely sacrificed.”

Thank you again for your affirmation of the manuscript. We tried our best to improve the manuscript and made some changes to the manuscript. These changes will not influence the core content and framework of the paper.

We sincerely appreciate for your thorough and insightful work, and hope that the corrections will meet with approval.

Once again, thank you very much for your comments and suggestions.

Best regards.

Yours sincerely,

Sen Zhao

References

  1. Kudou, K.; Komatsu, T.; Nogami, J.; Maehara, K.; Harada, A.; Saeki, H.; Oki, E.; Maehara, Y.; Ohkawa, Y. The requirement of Mettl3-promoted MyoD mRNA maintenance in proliferative myoblasts for skeletal muscle differentiation. Open Biol 2017, 7, https://doi.org/10.1098/rsob.170119.
  2. Yang, X.; Mei, C.; Raza, S. H. A.; Ma, X.; Wang, J.; Du, J.; Zan, L. Interactive regulation of DNA demethylase gene TET1 and m(6)A methyltransferase gene METTL3 in myoblast differentiation. International journal of biological macromolecules 2022, 223, 916-930, https://doi.org/10.1016/j.ijbiomac.2022.11.081.
  3. Petrosino, J. M.; Hinger, S. A.; Golubeva, V. A.; Barajas, J. M.; Dorn, L. E.; Iyer, C. C.; Sun, H.-L.; Arnold, W. D.; He, C.; Accornero, F. The m6A methyltransferase METTL3 regulates muscle maintenance and growth in mice. Nat Commun 2022, 13, 168, https://doi.org/10.1038/s41467-021-27848-7.
  4. Yang, X.; Mei, C.; Ma, X.; Du, J.; Wang, J.; Zan, L. m6A Methylases Regulate Myoblast Proliferation, Apoptosis and Differentiation. Animals : an open access journal from MDPI 2022, 12, https://doi.org/10.3390/ani12060773.
  5. Chen, J.; Niu, Y.; Wang, J.; Yang, Z.; Cai, Z.; Dao, X.; Wang, C.; Wang, Y.; Lin, Y. Physicochemical property, bacterial diversity, and volatile profile during ripening of naturally fermented dry mutton sausage produced from Jianzhou big-eared goat. Front Microbiol 2022, 13, 961117, https://doi.org/10.3389/fmicb.2022.961117.
  6. Zheng, S.; Li, L.; Zhou, H.; Zhang, X.; Xu, X.; Dai, D.; Zhan, S.; Cao, J.; Guo, J.; Zhong, T., et al. CircTCF4 Suppresses Proliferation and Differentiation of Goat Skeletal Muscle Satellite Cells Independent from AGO2 Binding. Int J Mol Sci 2022, 23, https://doi.org/10.3390/ijms232112868.

Round 2

Reviewer 2 Report

None

Author Response

We sincerely thank you for your comments, and we have made changes to the spelling and grammar of the article. Thank you again for your comments and suggestions.

Reviewer 3 Report

The manuscript has been totally improved. Minor spell check might be required.

Author Response

Dear Reviewer:

Thank you again for your comments concerning our manuscript entitled “METTL3 promotes the differentiation of goat skeletal muscle satellite cells by regulating MEF2C mRNA stability in m6A-dependent manner” (Manuscript ID: ijms-2576096). Your opinion is very insightful and helpful to the revision and improvement of our paper. We have carefully checked and revised the spelling of the article. The revised portions are marked in blue in this version of the paper. The main corrections in the paper in response to the reviewers’ comments are as following:

Line 18: “Nevertheless, it remains unclear how m6A participates in the development of goat muscle.”

l Line 21: “In addition, the global m6A modification level and differentiation of skeletal muscle satellite cells (MuSCs) were regulated by METTL3.”

Line 25: “Further experiments demonstrated that METTL3 enhanced the differentiation of MuSCs by up-regulating m6A levels and expression of MEF2C.”

Line 45: “and even involved with repair and regeneration after muscle injury.”

Line 89: “We performed qPCR to explore the spatio-temporal expression profile of METTL3 in goat tissues.”

Line 139: “The formation of myotubes was detected using the MyHC (red) immunofluorescence analysis in pEGFP-METTL3-transfected cells.”

Line 149: “Among these, six genes were randomly selected to evaluate the accuracy of the mRNA-seq data, which was confirmed by performing qPCR analysis.”

Line 159: “The formation of myotubes was detected using the MyHC (red) immunofluorescence analysis in pEGFP-METTL3-transfected cells.”

Line 188: “Furthermore, we overexpressed MEF2C in siMETTL3-transfected cells.”

Line 255: “Our study found a similar trend in that the expression of METTL3 of goat MuSCs gradually increased with the time of muscle differentiation.”

Line 342: “Plasmids and siRNAs were transfected into MuSCs using Lipofectamine 3000”

Line 414: “Protein interaction networks were constructed by String tool (https://cn.string-db.org) and Cytoscape software.”

We sincerely appreciate for your thorough and insightful work, and hope that the corrections will meet with approval. Once again, thank you very much for your comments and suggestions.

Best regards.

Yours sincerely,

Sen Zhao
